# The Impact of Neoadjuvant Chemotherapy on Survival Outcomes in Gastric Signet-Ring Cell Carcinoma: An International Multicenter Study

**DOI:** 10.3390/cancers17152419

**Published:** 2025-07-22

**Authors:** Yujuan Jiang, Peng Wang, Yantao Tian

**Affiliations:** Department of Pancreatic and Gastric Surgery, National Cancer Center/National Clinical Research Center for Cancer/Cancer Hospital, Chinese Academy of Medical Sciences and Peking Union Medical College, Beijing 100021, China; jyujuan@163.com (Y.J.); 13455863635@163.com (P.W.)

**Keywords:** gastric signet ring cell carcinoma, neoadjuvant chemotherapy, overall survival

## Abstract

This retrospective study analyzed 1773 SEER and 1289 NCC patients with gastric signet-ring cell carcinoma (GSRCC) undergoing radical surgery (2011–2018) to assess neoadjuvant chemotherapy (NAC) efficacy. NAC utilization was low (24.6% SEER, 22.6% NCC), with a median 30-month follow-up. Multivariate analyses identified tumor stage/size as survival predictors (*p* < 0.05), but NAC showed no overall survival benefit (SEER *p* = 0.653; NCC *p* = 0.139). Subgroup analysis revealed significant survival improvement with NAC in mid/distal tumors and cTNM II/III stages (all *p* < 0.001). While NAC demonstrated limited efficacy in unselected GSRCC, tailored use for locally advanced or anatomically specific cases may enhance outcomes. Further studies are warranted to optimize NAC strategies for this aggressive malignancy.

## 1. Introduction

Gastric cancer is the fifth most common malignancy globally, with over 1 million new cases diagnosed annually [1,2,3,4]. Radical surgical resection, often combined with multimodal therapies, remains the only curative treatment for this disease. Among the various subtypes of gastric adenocarcinoma, gastric signet-ring cell carcinoma (GSRCC) is characterized by the histological accumulation of cytoplasmic mucin that displaces the nucleus. GSRCC accounts for 14–35% of all gastric cancer cases worldwide [3,5,6]. Clinically, GSRCC is more aggressive than non-signet-ring cell carcinomas (non-SRCC), with 60–80% of patients presenting at advanced stages. The 5-year survival rate for locally advanced GSRCC is less than 20%, highlighting the critical need for effective therapeutic strategies [7,8,9].

The role of neoadjuvant chemotherapy (NAC) in GSRCC remains contentious, with conflicting evidence complicating consensus. While NAC has demonstrated survival benefits in esophagogastric adenocarcinoma [10], its efficacy in GSRCC remains uncertain, primarily due to the lack of histology-specific analyses in major clinical trials. For example, the MAGIC trial—a phase III study that established perioperative chemotherapy as the European standard for esophagogastric adenocarcinoma—excluded GSRCC from subgroup analyses [11]. A German multicenter study by Heger et al. further highlighted this uncertainty. Analyzing 723 locally advanced esophagogastric adenocarcinomas (including 32.5% with signet ring cell histology), they found that GSRCC patients exhibited significantly lower clinical (21.1% vs. 33.7%, *p* = 0.001) and histopathological response rates (16.3% vs. 28.9%, *p* < 0.001) to NAC compared to non-SRCC, along with worse median survival (26.3 vs. 46.6 months, *p* < 0.001). Notably, responders to NAC within the GSRCC subgroup still achieved improved survival (*p* = 0.003), suggesting that chemotherapy should not be universally abandoned in this population [12]. Emerging evidence suggests that intrinsic chemoresistance may contribute to the limited efficacy of NAC in GSRCC. A French nationwide cohort study (n = 1050) found that perioperative chemotherapy was associated with worse overall survival (OS) compared to surgery alone (hazard ratio [HR] = 1.40, 95% confidence intervals [95% CI]: 1.1–1.9, *p* = 0.042) [13]. These results are consistent with several retrospective studies [14,15,16], which have indicated no survival benefit from NAC in GSRCC patients. In contrast, a Dutch national registry study (n = 2046) reported a reduced risk of all-cause mortality within 90 days postoperatively for NAC recipients (HR = 0.29, 95% CI: 0.20–0.44; *p* < 0.001) [17].

Consequently, current clinical guidelines reflect this uncertainty. While the European Society for Medical Oncology (ESMO) conditionally recommends NAC for diffuse-type adenocarcinoma, they stress the need for histology-specific trials to better guide treatment decisions [18,19]. To address this gap, the PRODIGE 19 trial (NCT01717924) was launched in 2012 to compare perioperative chemotherapy with adjuvant therapy in GSRCC, though enrollment remains ongoing [20].

Despite these efforts, robust multicenter evidence on the survival benefits of NAC in GSRCC remains scarce. This study aims to evaluate the impact of NAC on survival in a large, international, multicenter cohort of GSRCC patients, providing high-quality evidence to guide clinical decision-making.

## 2. Materials and Methods

### 2.1. Study Population and Study Outcomes

Patients diagnosed with GSRCC between 2011 and 2018 were identified from the Surveillance, Epidemiology, and End Results (SEER) database (SEER*Stat v8.4.2) and the National Cancer Center (NCC) database. The inclusion criteria were as follows: (1) histology code 8490 (ICD-O-3), (2) primary tumor site (C16.0–C16.9), (3) no synchronous malignancies, (4) age between 20 and 80 years, and (5) receipt of radical surgery. Exclusion criteria included the following: (1) non-radical surgery, (2) unknown survival status, (3) distant metastasis, and (4) incomplete clinicopathological data (e.g., undetermined TNM stage). The screening process is outlined in Figure 1. Ethical approval for this study was granted by the Ethics Committee of the National Cancer Center, Chinese Academy of Medical Sciences (Approval No. 23/238-3980). Informed consent was waived due to the retrospective design of the study.

### 2.2. Diagnosis and Treatment

Pretreatment investigations included a physical examination, standard laboratory tests (including CEA, CA19-9, and CA72-4), a digestive endoscopy with biopsies and a computed tomography (CT) of the thorax, mediastinum, and abdomen. A total of 14.7% (189/1289) of patients underwent endoscopic ultrasound, predominantly those with early-stage disease (Stage I: 82.5%, n = 156; Stage II: 17.5%, n = 33). Patients in this study did not undergo diagnostic laparoscopy. GSRCC cases were strictly defined as tumors with >90% signet-ring cell composition, addressing variability in historical diagnostic thresholds (e.g., WHO criteria range from >50% to 90%).

Treatment strategies were determined by a multidisciplinary team, consisting of both medical and surgical oncologists and tailored to individual patient preferences. Patients were divided into two groups as follows: the upfront surgery group (surgery without neoadjuvant chemotherapy) and neoadjuvant chemotherapy (NAC) group. The NAC group received preoperative chemotherapy, followed by radical gastrectomy with standard D2 lymphadenectomy with or without adjuvant chemotherapy per guidelines. One of the following NAC regimens was used: SOX, FOLFOX, DOS, FLOT, or CAPEOX (Appendix A). The surgical approach (subtotal vs. total gastrectomy) was based on tumor location, with postoperative gastrointestinal reconstruction adjusted according to the type of resection. Resected specimens underwent centralized pathological review, and TNM staging was performed according to the AJCC 7th edition criteria. In our center, adjuvant chemotherapy was recommended for II/III stage patients. Adjuvant chemotherapy commenced 4–8 weeks postoperatively and was administered for 2–4 cycles. For patients receiving neoadjuvant therapy, NAC protocols are generally recommended as an adjuvant chemotherapy regimen.

### 2.3. Follow-Up and Statistical Analysis

Patients were followed-up at regular intervals, as follows: every 3 months for the first 2 years and biannually thereafter until 5 years post-surgery. Follow-up evaluations included tumor marker testing, chest-abdominal-pelvic CT scans, and annual endoscopic examinations. The primary endpoint of the study was OS, defined as the time from treatment initiation to death or last follow-up. The frequency of missing data for patient-, tumor-, and treatment-related variables was also recorded. Continuous variables were expressed as the mean ± standard deviation and were analyzed by a t-test for continuous variables. Categorical variables were displayed as counts and percentages, and they were analyzed using χ^2^ tests or Fisher’s exact tests. Survival analysis was performed using Kaplan–Meier curves and Cox proportional hazards models (both univariate and multivariate). Subgroup comparisons between the NAC and surgery-alone groups were performed, with statistical significance set at *p* < 0.05 (two-tailed). Censoring occurred for patients who remained alive without relapse at their last follow-up visit. All statistical analyses were conducted using R version 3.5.1. Detailed R package information are showed in Appendix A.

## 3. Results

### 3.1. Study Population

This study included 3062 patients diagnosed with GSRCC between 2011 and 2018, comprising two cohorts as follows: 1289 patients from the NCC database and 1773 patients from the SEER registry. Baseline characteristics of the study population are summarized in Table 1. The median age was 63 years (range: 20–80), with a male-to-female ratio of 1.44:1. Tumors were predominantly localized to the mid/distal stomach in 63.3% of cases, with 7.7% exhibiting overlapping lesions. At diagnosis, 65.4% of patients presented with locally advanced disease (cTNM stage II/III).

### 3.2. Treatment and Histopathological Outcomes

Treatment strategies are outlined in Table 2. NAC was administered to a relatively small proportion of patients as follows: 24.6% (436/1773) of patients in the SEER cohort and 22.6% (292/1289) in the NCC cohort. The majority of patients in the NCC database received NAC according to the SOX (n  =  170; 58.2%) or DOS protocols (n  =  77; 26.3%). The majority of patients received upfront surgery (SEER: 75.4%, NCC: 77.3%). Surgical data available for the NCC cohort revealed that total gastrectomy was the predominant procedure (59.9%, 772/1289), followed by subtotal resection in 33.3% (429/1289) of cases. Details of adjuvant therapy are shown in Appendix A. In the upfront surgery group, there are 558 patients in the SEER database and 304 patients in the NCC database who received adjuvant therapy. Among the 456 patients who received standard adjuvant chemotherapy in the NCC database, 88 (19.3%) received single-agent 5-fluorouracil, and 368 (80.7%) received multiagent chemotherapy (a combination of 5-fluorouracil and cisplatin/oxaliplatin, doxorubicin, or paclitaxel/docetaxel). Among the 3062 patients, 48.1% (n = 1472) underwent surgery without any perioperative therapy, reflecting the historical preference for upfront resection in GSRCC. This proportion varied between databases as follows: 43.9% (779/1773) in SEER and 53.8% (693/1289) in NCC, likely due to regional differences in treatment guidelines during the study period. Histopathological characteristics are presented in Table 3. Most tumors were classified as pT3–T4 (SEER: 55.6%, NCC: 57.7%) and pN3 (SEER: 22.1%, NCC: 29.0%). Poor differentiation was observed in 79.6% of tumors (2436/3062), with higher rates in the SEER cohort (84.0%, 1489/1773) compared to the NCC cohort (73.5%, 947/1289). Lymph node dissection counts varied between the cohorts: the SEER cohort averaged 27.4 ± 40.1 nodes (19.3 ± 21.5 metastatic nodes), while the NCC cohort averaged 32.0 ± 14.9 nodes (6.0 ± 9.7 metastatic nodes).

### 3.3. Outcome and Prognostic Factors

The median follow-up duration was 30 months (range: 8–131 months; IQR: 24–70). In the SEER cohort, the 3-year survival rate was 47.4%, and the 5-year survival rate was 41.3% (Figure 2A). In the NCC cohort, the 3-year survival rate was 82.4%, and the 5-year survival rate was 73.9% (Figure 2B). There were no significant differences in 5-year OS between the NAC and surgery-alone groups (SEER: 45.6% vs. 48.0%, *p* = 0.31; NCC: 79.0% vs. 82.2%, *p* = 0.96) (Figure 3). Among 305 patients with documented disease recurrence in the NCC cohort, 20.5% patients (60/305) experienced recurrence in the NAC group, while 24.5% (245/997) experienced recurrence in the upfront surgery group (χ^2^ = 2.02, *p* = 0.155; detailed in Appendix A). There was no significant difference in treatment between the two groups after relapse.

The number of patients at risk at each interval is shown in the table at the bottom of the graph.

Univariate analysis revealed several factors associated with poor prognosis in the SEER cohort, including race, tumor size, cTNM stage, pT, and pN stage (Table 4). In the NCC cohort, significant prognostic factors included age, tumor size, cTNM stage, NAC, and adjuvant chemotherapy. Multivariate analysis confirmed these factors as independent predictors of survival in SEER (all *p* < 0.05; Figure 4), while in the NCC cohort, age, tumor size, and cTNM stage remained significant (*p* < 0.05). NAC did not show a significant OS benefit in either cohort (SEER: *p* = 0.653; NCC: *p* = 0.139).

Subgroup analyses (Figure 5) indicated that NAC did not confer substantial survival benefits in the overall GSRCC cohort. However, in patients with mid/distal tumors and those with cTNM II/III tumors, the distribution of 95% CIs tended to focus on the NAC strategy (*p* < 0.001). Further survival analyses for these subgroups demonstrated that NAC significantly improved 3-year OS in GSRCC patients with tumors located in the distal stomach (*p* = 0.0097; Figure 6).

## 4. Discussion

The increasing incidence of GSRCC, particularly among younger populations in Western countries, alongside its aggressive biological behavior and worse prognosis compared to non-SRCC, underscores the urgent need for optimized therapeutic strategies [8,15,21,22]. Our multicenter study, incorporating data from the SEER registry and the NCC database (n = 3062), is the largest cohort to date investigating the role of NAC in GSRCC. While NAC did not demonstrate a 5-year OS benefit in the entire cohort (SEER: 45.6% vs. 48.0%, *p* = 0.31; NCC: 79.0% vs. 82.2%, *p* = 0.96), subgroup analyses indicated potential survival benefits for patients with mid/distal tumors or those with locally advanced tumors (cTNM II/III), suggesting that NAC may be effective in these specific patient populations.

Most major esophagogastric cancer trials on pre- or perioperative multimodal treatment regimens—such as the MAGIC, FFCD, and CROSS trials—have not included SRCC or diffuse gastric cancers [23,24]. The FLOT4 trial did offer a subgroup analysis comparing SRCC to non-SRCC, revealing a non-significant survival benefit of FLOT over an epirubicin-based perioperative chemotherapy regimen for SRCC patients [10]. To date, no phase III trials have specifically examined the efficacy of neoadjuvant therapy for GSRCC.

Our findings generally align with previous retrospective studies, such as the French national survey (n = 924), which showed comparable R0 resection rates (62.3% vs. 65.9%) between NAC and surgery-alone groups, but there was no significant benefit in median survival in the NAC cohort (12.8 vs. 14.0 months) [13]. Similarly, a meta-analysis of patients with esophagogastric signet-ring cell carcinoma undergoing NAC followed by surgery reported no significant survival benefit (HR 1.01; 95% CI, 0.61–1.67; *p* = 0.98) [25]. In our previous study, we detected significantly less advanced (y)pT for patients after NAC, but NAC did not improve the R0 resection rate or result in lymph node downstaging [16]. However, there is conflicting evidence, with some studies suggesting that NAC can prolong median OS, likely due to tumor downstaging and improved resectability [26], as observed by Heger et al. (28.5 vs. 14.9 months, *p* < 0.001). Furthermore, a study of 1728 diffuse gastric cancer patients reported a reduction in all-cause mortality within 90 days postoperatively after NAC (HR: 0.29, 95% CI: 0.20–0.44; *p* < 0.001) [17]. These discrepancies may arise from differences in patient selection, chemotherapy regimens (e.g., FLOT vs. ECF), and the inclusion of diverse anatomical subgroups.

One of the most compelling findings of our study was the significant racial disparity in survival outcomes among patients with GSRCC. Our multivariable analysis revealed that Chinese patients in the SEER cohort exhibited markedly better overall survival (HR: 0.55, 95% CI: 0.37–0.81, *p* = 0.002), even after adjusting for tumor stage, size, and treatment modalities. This aligns with the prior literature suggesting racial disparities in gastric cancer outcomes, potentially due to differences in tumor biology (e.g., higher prevalence of diffuse-type histology in Western populations), access to care, or socioeconomic factors [27]. While SEER’s Chinese subgroup exhibited improved survival—potentially linked to clinicopathological or socioeconomic factors—this association was absent in the pooled analysis. This discrepancy highlights the need for molecularly informed studies to disentangle genetic ancestry from sociocultural determinants of outcome. Our study suggests that NAC may benefit a subgroup of patients with mid/distal tumors, possibly due to better drug delivery enabled by anatomical features (e.g., vascularization and lymphatic drainage) [28]. While molecular differences (e.g., CDH1 mutations in distal tumors) may contribute to chemotherapy sensitivity [29,30], the clinical relevance of these findings requires validation in prospective studies. In addition, our finding suggests that the staging status of the tumor may be a key factor influencing the efficacy of NAC. Locally advanced tumors may exhibit distinct biological behaviors compared to early-stage tumors, including altered tumor cell proliferation, metabolism, and angiogenesis, which may impact their sensitivity and responsiveness to chemotherapy drugs. For example, locally advanced tumors may harbor more chemotherapy-resistant clones, and NAC could help reduce the risk of postoperative recurrence by targeting these resistant populations [28].

This study introduces several innovations in evaluating the efficacy of neoadjuvant therapy for GSRCC. By integrating data from both Eastern (NCC) and Western (SEER) populations, we have established the largest multicenter cohort, improving the reliability and statistical power of our findings. Our cross-regional comparative analysis allows for a broader understanding of treatment responses and clinical practices. Importantly, we identify mid/distal and locally advanced tumors as subsets of GSRCC patients who may benefit from NAC, paving the way for precision oncology in the management of this aggressive malignancy.

Patients undergoing non-radical surgery (R1/R2 resection) were excluded to minimize confounding by residual tumor burden, as incomplete resection independently predicts poor prognosis and may obscure the evaluation of NAC efficacy. However, it is important to acknowledge the clinical challenges posed by borderline resectable or metastatic disease, which were excluded from this analysis. GSRCC’s inherent biological aggressiveness—manifested as diffuse infiltration, peritoneal dissemination, and high rates of R1/R2 resection—underscores the urgent need to evaluate NAC’s role beyond curative-intent settings such as cytoreduction or symptom control. Future studies should prioritize prospective trials evaluating targeted therapies for advanced GSRCC, which may expand therapeutic options for patients currently deemed incurable.

This study has several limitations that warrant consideration. First, its retrospective cohort design inherently carries risks of selection bias. Despite rigorous adjustment for measurable confounders, residual selection bias may persist due to unmeasured factors such as detailed performance status, nutritional parameters, or molecular subtypes. Second, the SEER database lacks critical molecular profiling data, including HER2 expression, MSI/MMR status, and PD-L1 levels, which are essential for characterizing tumor biology and guiding targeted therapies. Similarly, the absence of MMR/MSI and PD-L1 data in the NCC database reflects historical constraints of the study period. The unavailability of molecular profiles precluded comprehensive analyses of predictive biomarkers for chemotherapy response, limiting insights into precision oncology strategies. Additionally, the SEER database does not provide granular details on perioperative treatment regimens (e.g., chemotherapy cycles, radiation protocols), hindering robust assessment of their prognostic impact on GSRCC. Finally, the combined use of the SEER and NCC databases aimed to maximize statistical power for subgroup analyses while evaluating the generalizability of NAC effects across ethnogeographic contexts. However, the racial and geographic heterogeneity between SEER (predominantly White, U.S.-based) and NCC (exclusively Asian, China-based) cohorts may introduce unmeasured confounding. Although we adjusted for race in multivariable models, residual confounding may persist.

## 5. Conclusions

In conclusion, our findings advocate for a precision oncology approach in GSRCC management. While NAC does not show substantial efficacy in unselected populations, its selective application in mid/distal or locally advanced tumors may offer significant survival benefits. Moving toward clinicopathologically guided therapy will be crucial in improving outcomes for this challenging and aggressive disease.

## Figures and Tables

**Figure 1 cancers-17-02419-f001:**
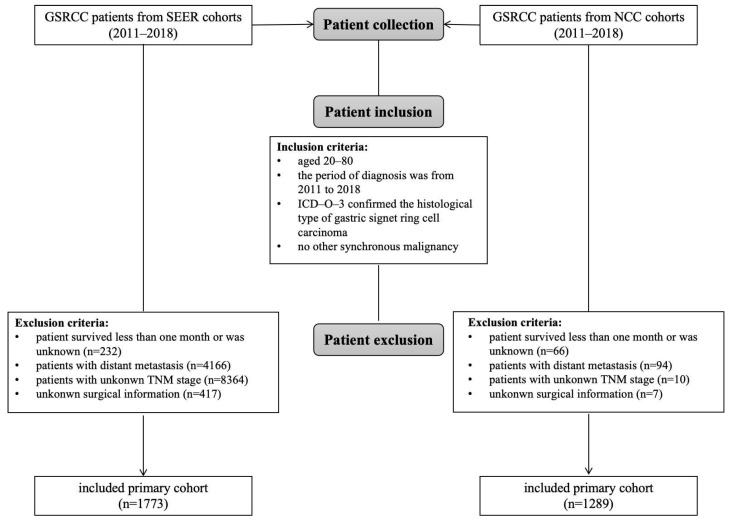
Flow diagram of the study.

**Figure 2 cancers-17-02419-f002:**
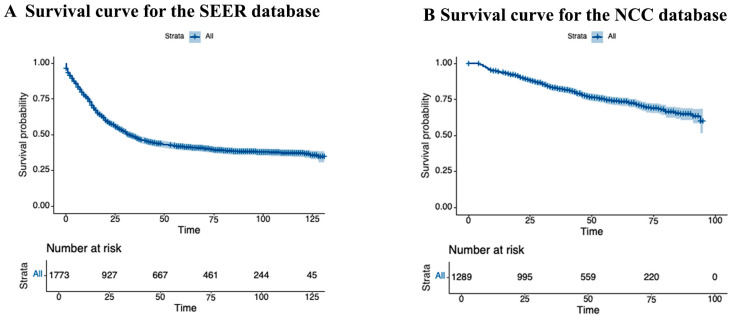
Overall survival of the entire cohort. (**A**) SEER database; (**B**) NCC database.

**Figure 3 cancers-17-02419-f003:**
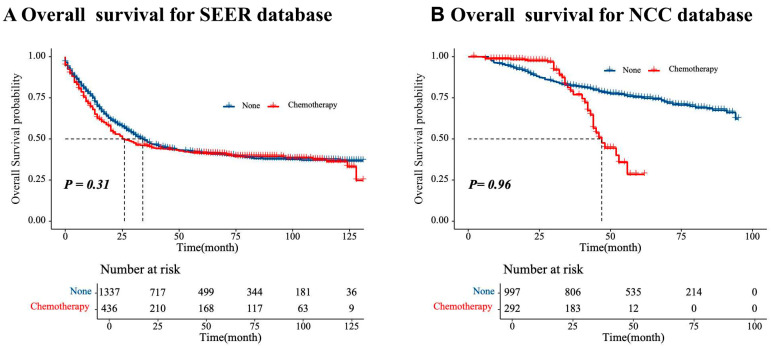
Survival curves for the NAC and upfront surgery groups. (**A**) SEER database; (**B**) NCC database.

**Figure 4 cancers-17-02419-f004:**
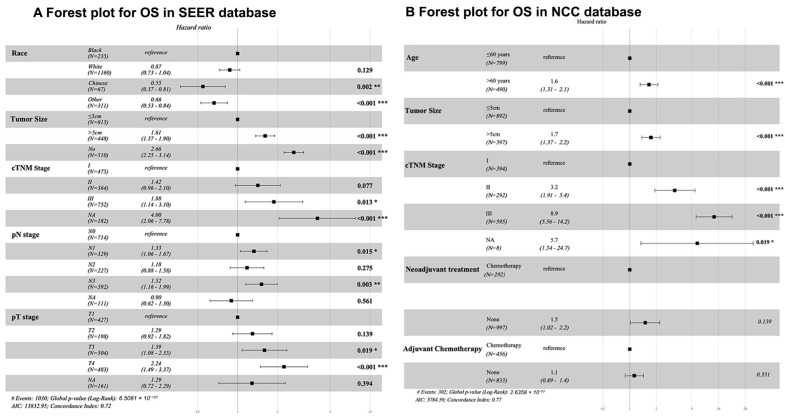
Forest plots from multivariate Cox regression analysis assessing overall survival in patients with GSRCC. (**A**) SEER database; (**B**) NCC database. * *p* < 0.05, ** *p* < 0.01, *** *p* < 0.001.

**Figure 5 cancers-17-02419-f005:**
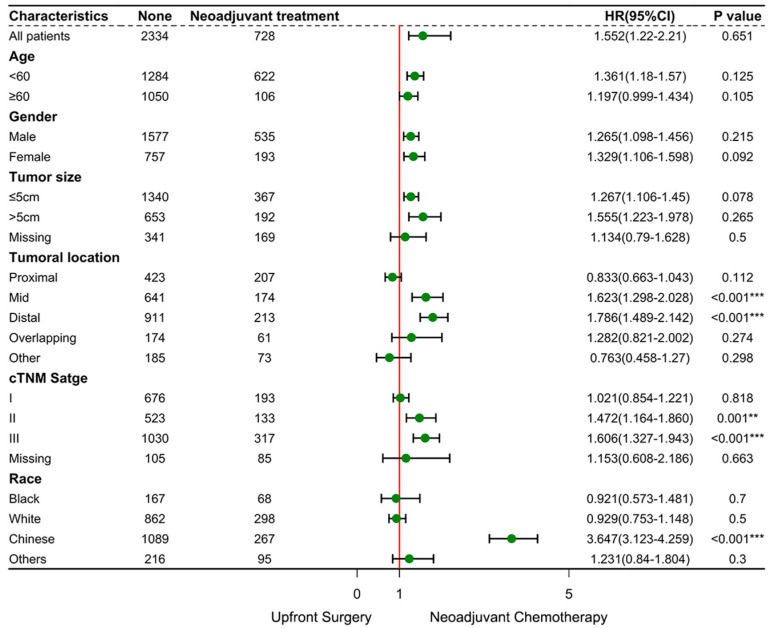
Treatment effects according to the baseline characteristics of the patients. The forest plot shows HRs for mortality (indicated by circles) and their corresponding 95% CIs (I bars). *p*-values reflect the statistical interaction between treatment modalities and each subgroup variable. ** *p* < 0.01, *** *p* < 0.001. HR = hazard ratio. CI = confidence interval.

**Figure 6 cancers-17-02419-f006:**
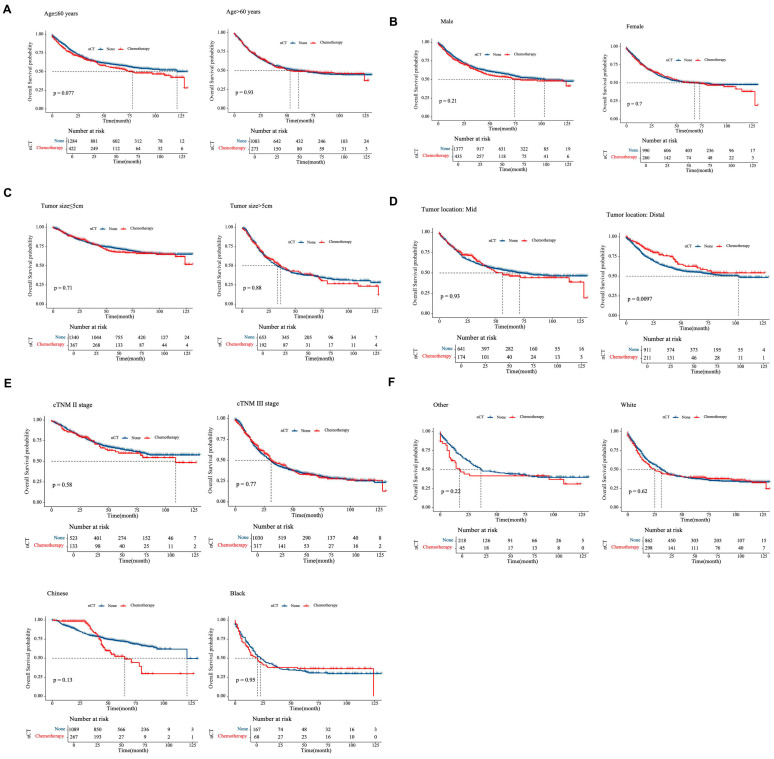
Overall survival curves for GSRCC patients, stratified by clinical factors. (**A**) Age. (**B**) Gender. (**C**) Tumor size. (**D**) Tumor location. (**E**) cTNM stage. (**F**) Race. The number of patients at risk at each interval is shown in the table at the bottom of the graph.

**Table 1 cancers-17-02419-t001:** Baseline characteristics of patients who were diagnosed with GSRCC and who underwent surgical resection.

Variables	SEER Database (N = 1773, %)	NCC Database (N = 1289, %)	*p*-Value	Total (N = 3062, %)
Gender			0.233	
Male	1238 (69.8%)	874 (67.8%)		2112 (69.0)
Female	535 (30.2%)	415 (32.2%)		950 (31.0)
Age, years			0.800	
≤60	1107 (62.4%)	799 (62.0%)		1906 (62.2)
>60	666 (37.6%)	490 (38.0%)		1156 (37.8)
Race			<0.001	
Black	235 (13.3%)	0		235 (7.7)
White	1160 (65.4%)	0		1160 (37.9)
Chinese	67 (3.8%)	1289 (100%)		1356 (44.3)
Other	311 (17.5%)	0		311 (10.2)
Tumoral locations			<0.001	
Proximal	316 (17.8%)	314 (24.4%)		630 (20.6)
Mid	506 (28.5%)	309 (24.0%)		815 (26.6)
Distal	528 (29.8%)	596 (46.2%)		1124 (36.7)
Overlapping	165 (9.3%)	70 (5.4%)		235 (7.7)
Unknown	258 (14.6%)	0		258 (8.4)
Tumor Size, cm			<0.001	
≤5	815 (46.0%)	892 (69.2%)		1707 (55.7)
>5	448 (25.3%)	397 (30.8%)		845 (27.6)
NA	510 (28.8%)	0		510 (16.6)
Year of surgery			0.227	
2011–2014	812 (45.8%)	562 (43.6%)		1374 (44.8)
2015–2018	961 (54.2%)	727 (56.4%)		1688 (55.2)
Pretherapeutic cTNM stage			0.003	
I	475 (26.8%)	394 (30.6%)		869 (28.3)
II	364 (20.5%)	292 (22.7%)		656 (21.4)
III	752 (42.4%)	595 (46.2%)		1347 (44.0)
NA	182 (10.3%)	8 (0.6%)		190 (6.2)

The proximal region includes the fundus; the mid region includes the corpus, lesser curvature, and greater curvature; the distal region includes the antrum and pylorus. Percentages may not add up to 100% due to rounding.

**Table 2 cancers-17-02419-t002:** Treatment characteristics of 3062 resected patients.

Variables	SEER Database (n = 1773, %)	NCC Database (n = 1289, %)	*p*-Value	Total (n = 3062, %)
Neoadjuvant treatment			0.159	
Chemotherapy	436 (24.6%)	292 (22.6%)		728 (23.8)
None	1337 (75.4%)	997 (77.3%)		2334 (76.2)
Adjuvant treatment			<0.001	
Chemotherapy	296 (16.7%)	456 (35.4%)		752 (24.6)
Chemoradiotherapy	388 (21.9%)	0		388 (12.7)
Radiotherapy alone	54 (3.0%)	0		54 (1.7)
None	1035 (58.4%)	833 (64.6%)		1868 (61.0)
Surgical procedure a			**-**	
Extended resection to neighboring organs	-	3 (0.2%)		
Total gastrectomy	-	772 (59.9)		
Subtotal gastrectomy	-	429 (33.3)		
Unknown type of procedure	-	88 (6.8)		

Percentages may not add up to 100% due to rounding. a Descriptions of surgical modalities are lacking in the SEER database.

**Table 3 cancers-17-02419-t003:** Histopathological outcomes of 3062 resected patients.

Variables	SEER Database (n = 1773, %)	NCC Database (n = 1289, %)	*p*-Value	Total (n = 3062, %)
Tumor stage				
pT			0.021	
Tis	0	3 (0.2)		3 (0.1)
T1	427 (24.1)	384 (29.8)		811 (26.5)
T2	198 (11.2)	157 (12.2)		355 (11.6)
T3	504 (28.4)	295 (22.9)		799 (26.1)
T4	483 (27.2)	449 (34.8)		932 (30.4)
Unknown	161 (9.1)	1 (0.1)		162 (5.3)
pN			<0.001	
N0	714 (40.3)	537 (41.7)		1251 (40.9)
N1	329 (18.6)	179 (13.9)		508 (16.6)
N2	227 (12.8)	192 (14.9)		419 (13.7)
N3	392 (22.1)	374 (29.0)		766 (25.0)
Unknown	111 (6.3)	7 (0.5)		118 (3.8)
Tumor differentiation			<0.001	
Well	3 (0.2)	9 (0.7)		12 (0.4)
Moderate	35 (2.0)	259 (20.1)		294 (9.6)
Poorly	1489 (84)	947 (73.5)		2436 (79.6)
Undifferentiated	41 (2.3)	1 (0.1)		42 (1.3)
Unknown	205 (11.6)	73 (5.7)		278 (9.1)
HER2 ^a^	-			
Loss of expression	-	593 (46.0)		
Low expression	-	497 (38.6)		
Overexpression	-	189 (14.7)		
Not detected	-	10 (0.7)		
Number of dissected lymph nodes [mean ± SD]	27.4 ± 40.1	32.0 ± 14.9	<0.001	-
Number of invaded lymph nodes [mean ± SD]	19.3 ± 21.5	6.0 ± 9.7	<0.001	-

^a^ Due to the limitations of the seer database, molecular data such as HER2 status are usually not included in the SEER database.

**Table 4 cancers-17-02419-t004:** Univariable and multivariable Cox regression analyses on the influence of NAC on overall survival for GSRCC.

Variables	SEER Database				NCC Database			
	Univariable Analysis		Multivariable Analysis		Univariable Analysis		Multivariable Analysis	
	HR (95% CI)	*p*-Value	HR (95% CI)	*p*-Value	HR (95% CI)	*p*-Value	HR (95% CI)	*p*-Value
**Sex**								
Male	REF				REF			
Female	1.07 (0.95–1.21)	0.253			0.84 (0.66–1.08)	0.185		
**Age (year)**								
<60	REF				REF			
>60	0.94 (0.83–1.07)	0.348			2.02 (1.61–2.54)	<0.001	1.65 (1.31–2.07)	**<0.001**
**Race**								
Black	REF		REF					
White	0.86 (0.72–1.02)	0.082	0.87 (0.73–1.04)	0.129	-	-	-	-
Chinese	0.53 (0.36–0.77)	0.001	0.55 (0.37–0.81)	**0.002**	-	-	-	-
Others	0.56 (0.5–0.70)	<0.001	0.66 (0.53–0.84)	**<0.001**	-	-	-	-
**Tumor size (cm)**					-	-	-	-
<5	REF		REF		REF			
>5	2.69 (2.31–3.13)	<0.001	1.61 (1.37–1.90)	**<0.001**	3.09 (2.46–3.87)	<0.001	1.73 (1.37–2.20)	**<0.001**
Unknown	3.18 (2.74–3.7)	<0.001	2.66 (2.25–3.14)	**<0.001**	-	-	-	-
**Tumor location**								
Proximal	REF				REF			
Mid	1.07 (0.89–1.29)	0.462			0.81(0.72–0.93)	0.425		
Distal	1.1 (0.91–1.32)	0.318			0.15 (0.05–0.35)	0.150		
Overlapping	1.06 (0.82–1.35)	0.670			0.22 (0.20–5.53)	0.980		
Unknown	1.02 (0.82–1.27)	0.872			1.25 (0.15–3.69)	0.115		
**Histological grade**								
Well	REF				REFREF			
Moderate	1.69 (0.23–12.71)	0.610			4.22 (1.23–9.88)	0.991		
Poorly	2.07 (0.29–14.7)	0.468			6.39 (2.56–17.21)	0.991		
Undifferentiated	1.83 (0.25–13.53)	0.555			4.81 (4.10–8.20)	0.989		
Unknown	2.04 (0.29–14.62)	0.477			2.79 (0.55–7.88)	0.991		
**pT stage**								
T1	REF		REF		REF			
T2	1.47 (1.11–1.95)	0.007	1.29 (0.92–1.82)	0.139	0.2 (0.03–1.51)	0.119		
T3	2.81 (2.28–3.47)	<0.001	1.59 (1.08–2.35)	**0.019**	0.47 (0.06–3.50)	0.459		
T4	4.82 (3.93–5.92)	<0.001	2.24 (1.49–3.37)	**<0.001**	1.5 (0.21–10.75)	0.688		
Unknown	7.19 (5.58–9.27)	<0.001	1.29 (0.72–2.29)	0.394	2.32 (0.32–16.58)	0.402		
**p** **N stage**								
N0	REF		REF		REF			
N1	1.94 (1.62–2.32)	<0.001	1.33 (1.06–1.67)	**0.015**	1.11 (0.08–15.21)	0.552		
N2	2.06 (1.69–2.51)	<0.001	1.18 (0.88–1.58)	0.275	3.36 (0.55–7.55)	0.152		
N3	3.21 (2.73–3.79)	<0.001	1.52 (1.16–1.99)	**0.00** **3**	0.05 (0.01–2.55)	0.335		
Unknown	4.43 (3.47–5.67)	<0.001	0.90 (0.62–1.30)	0.561	5.56 (1.12–7.88)	0.782		
**cTNM stage**								
I	REF				REF		REF	
II	2.15 (1.71–2.70)	<0.001	1.42 (0.96–2.10)	0.077	3.31 (1.99–5.53)	<0.001	3.20 (1.91–5.40)	**<0.001**
III	4.69 (3.86–5.70)	<0.001	1.88 (1.14–3.20)	**0.013**	11.06 (7.07–17.31)	<0.001	8.90 (5.56–14.20)	**<0.001**
Unknown	7.67 (6.01–9.80)	<0.001	4.00(2.06–7.78)	**<0.001**	6.09 (1.43–26.02)	0.015	5.70 (1.34–24.70)	**0.01** **9**
**Year of surgery**								
2011–2014	REF				REF			
2015–2018	0.97 (0.86–1.10)	0.683			0.82 (0.64–1.04)	0.107		
**Neoadjuvant treatment**								
Chemotherapy	REF				REF		REF	
None	0.96 (0.81–1.14)	0.653			1.40 (0.96–2.05)	0.08	1.49 (1.02–2.18)	0.139
**Adjuvant chemotherapy**								
Presence	REF				REF		REF	
Absence	0.94 (0.72–1.23)	0.638			1.40 (0.96–2.05)	0.08	1.10 (0.89–1.40)	0.331
Unknown	0.91 (0.78–1.07)	0.257			-	-	-	-

## Data Availability

The data that support the findings of this study are available from the corresponding author upon reasonable request.

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
