# Peer review of "The Impact of Neoadjuvant Chemotherapy on Survival Outcomes in Gastric Signet-Ring Cell Carcinoma: An International Multicenter Study"

_cancers, 2025, doi:10.3390/cancers17152419_

Round 1

Reviewer 1 Report

Comments and Suggestions for Authors

This retrospective analysis by Yujuan Jiang et al includes a large cohort of 3062 patients with gastric signet-ring cell carcinoma (GSRCC) with data from both Eastern and Western populations undergoing radical surgery to assess neoadjuvant chemotherapy (NAC) efficacy. The authors should be congratulated, since the paper is well written and aims to assess a clinical unmet need in the management of theses patients, the role of NAC in this particular histological subtype. In my opinion, some issues need however attention;

1- What type of NAC did the patient receive? This might be importans since, as the authors mention in the Discussion, a non significant survival benefit has been reported with the use of taxane-containing combinations in GSRCC patients

2-Some information about the molecular profile is missing, at least the presence of MSI, expression of Her-2 or PDL-1. MSI or PDL-1 expression are found in about one third of advanced GSRCC patients (Jin S et al, Oncotarget 2017 21; 8) and may have influenced outcome

3- How many patients underwent post-relapse therapy. And what type? Was it well balanced between NAC and Surgery-only groups? This may be particularly relevant with the postrelapse use of immunotherapy, which may translate into prolongued survival times in certain subsets of patients

4-Do patients in the Surgery-Only group receive any type of adjuvant therapy?

5-Did the initial work-up of these patients include an upper endoscopic ultrasound or a diagnostic laparoscopy? How often?

Reviewer 2 Report

Comments and Suggestions for Authors

This is valuable retrospective study of large population of gastric signet-ring cell carcinoma patients (both American and Chinese) undergoing radical surgery prior FLOT era (before 2019) of neoadjuvant chemotherapy (NAC), that was utilized only in about 24%. Multivariate analyses identified tumour stage as survival predictor, but NAC showed no overall survival benefit, except in mid/distal tumours and cTNM II/III stages by subgroup analysis. Authors concluded that tailored use for locally advanced or anatomically specific cases may enhance outcomes.

I have some important comments that could be used to further improve the manuscript.

  1. Introduction: no mention on German study by Heger et al. from 2014 (DOI 10.1245/s10434-013-3462-z)
  2. Materials & Methods: non-radical surgery as exclusion criterion, please discuss the issue. It is well known that the SRC gastric cancer is risk factor for non-radical surgery.
  3. Combination of the two data bases: American SEER (predominantly White race 65%) and Chinese NCC. Could you please justify combined use of such a heterogeneous population.
  4. More detailed description of the statistical software package (R version 3.5.1.)
  5. Table 2: 76% of pts have not received any therapy before resection, while only 39% have received an adjuvant therapy. What proportion of pts has received neither neo-, nor adjuvant therapy (surgery alone)?
  6. Please add statistical comparison of the SEER and NCC groups in Tables 1-3. This is esp. important in Table 3 (pTNM), since survival in both cohorts is dramatically different (Figure 2).
  7. Chinese race emerged as significant prognostic factor in the SEER only, and why not in the entire cohort? (Figure & Table 4)
  8. Therefore, further subgroup analysis (Figures 5 & 6) excluding the race factor is not justified.
  9. No discussion on the race as significant prognostic factor, which is the most interesting finding of the present study. Please shorten the paragraph on molecular disparities (lines 232-244) since it is somewhat speculative.
  10. This study is retrospective and rely on database analyses, which come with inherent limitations. For example, although significant survival benefits have been demonstrated for patients with mid/distal or those with locally advanced tumours, there is a potential for selection bias. Patients with better physical status and fewer comorbidities are more likely to be chosen for neoadjuvant chemotherapy, potentially leading to improved survival outcomes.

Round 2

Reviewer 2 Report

Comments and Suggestions for Authors

I am satisfied with the corrections made to the manuscript.